# Luteolin target HSPB1 regulates endothelial cell ferroptosis to protect against radiation vascular injury

Li Wen[1], Weiyuan Zhang[2,3], Jia Hu[4], Tao Chen[4], Yiming Wang[1], Changchang Lv[1], Min Li[5]*, Lisheng Wang[1,3]*, Fengjun Xiao[5]*

1 School of Nursing, Jilin University, Changchun, P. R. China, 2 Department of Special Medicine, School of Basic Medicine, Qingdao University, Qingdao, P. R. China, 3 Laboratory of Molecular Diagnosis and Regenerative Medicine, The Affiliated Hospital of Qingdao University, Qingdao, P. R. China, 4 Department of Cardiovascular, The Sixth Medical Center of Chinese PLA General Hospital, Haidian District, Beijing, China, 5 Beijing Institute of Radiation Medicine, Beijing, P. R. China

* limin82057@163.com (ML); lishengwang@jlu.edu.cn (LW); xiaofjun1105@163.com (FX)

## Abstract

Vascular endothelial damage due to ionizing radiation is the main pathological process of radiation injury and the main cause of damage to various organs in nuclear accidents. Ferroptosis plays an important role in ionizing radiation-induced cell death. We have previously reported that luteolin is highly resistant to ferroptosis. In the present study, body weight, microvessel count, H&E, and Masson staining results showed that luteolin rescued radial vascular injury *in vivo*. Cell Counting Kit 8 (CCK8), Giemsa staining clarified the anti-ferroptosis ability of luteolin with low toxicity. Malondialdehyde (MDA), superoxide dismutase (SOD), NADP$^+$/NADPH, Fe$^{2+}$ staining, dihydroethidium (DHE) and MitoTracker assays for ferroptosis-related metrics, we found that luteolin enhances human umbilical vein endothelial cells (HUVECs) antioxidant damage capacity. Drug affinity responsive target stability (DARTS), surface plasmon resonance (SPR), computer simulated docking and western blot showed that heat shock protein beta-1 (HSPB1) is one of the targets of luteolin action. Luteolin inhibits ferroptosis by promoting the protein expression of HSPB1/solute carrier family 7 member 11 (SLC7A11)/ glutathione peroxidase 4 (GPX4). In conclusion, we have preliminarily elucidated the antioxidant damage ferroptosis ability and the target of action of luteolin to provide a theoretical basis for the application of luteolin in radiation injury diseases.

**Data Availability Statement:** All relevant data are within the manuscript and its Supporting Information files.

## 1. Introduction

Ionizing radiation (IR), such as nuclear accidents, nuclear terrorist attacks and radiotherapy, can cause radioactive damage to tissues [1–4]. In particular, radiotherapy, as the mainstay of medical oncology treatment, causes all kinds of unavoidable complications, and radiation-induced endothelial/vascular damage is one of the main complications, which is also a major cause of morbidity and mortality in nuclear or radiological disasters [5]. Vascular endothelial

**Funding:** This research was funded by the National Natural Science Foundation of China, grant number 82073489.

**Competing interests:** The authors have declared that no competing interests exist.

cells play an important role in the circulatory system as the single-cell lining of the heart and all blood vessels. It is in direct contact with chemicals or particles in the circulatory system and maintains multi-organ health and homeostasis by controlling solute permeability, sensing shear stress, dynamically maintaining vasodilatory tone, and playing a role in anti-inflammatory and pro-inflammatory, antioxidant, and pro-oxidant properties [6,7]. When cells are exposed to IR, a stress response occurs in less than a microsecond, which interferes, affecting all organelles and their molecular mechanisms through the formation of reactive oxygen interactions with biological substances such as DNA, proteins and lipids [8,9]. Radiation causes endothelial cell swelling, necrosis and related inflammatory responses, which in turn cause pathological changes such as tissue necrosis and ischemia, which become the cause of its long-term complications [10,11]. Due to the relatively small number of clinical cases, the current development of radioprotective drugs remains mainly at the basic research stage, and large-scale clinical application is still a long way off [12]. It is therefore essential to discover protective drugs for radiological vascular injury.

Research indicates that ferroptosis plays a crucial role in radiation-induced cell death responses. Radiation can cause tumor cells to generate a significant amount of lipid reactive oxygen species (ROS), leading to the accumulation of lipid peroxides and subsequently inducing ferroptosis. Ionizing radiation can produce hydroxyl radicals within the lipid bilayer and promote lipid peroxidation [13]. The use of ferroptosis inducers to inhibit solute carrier family 7 member 11 (SLC7A11) or glutathione peroxidase 4 (GPX4) activity can enhance the sensitivity of tumor cells to radiotherapy (RT) [14,15]. Studies have shown that RT significantly increases the staining of C11-BODIPY and the lipid peroxidation marker malondialdehyde (MDA) in cancer cells and tumor samples. Post-irradiation, cells exhibit morphological characteristics of ferroptosis, including mitochondrial shrinkage and increased membrane density. Ferroptosis inhibitors such as ferrostatin-1 (fer-1) or deferoxamine (DFO) can partially restore the clonogenic survival of various cancer cell lines after RT [14,16]. Mechanistically, IR induces lipid peroxidation and ferroptosis through at least three pathways [14,16,17]. Firstly, IR can induce lipid peroxidation by generating excess ROS. ROS produced by IR can extract electrons from polyunsaturated fatty acids (PUFAs), leading to the formation of PUFA radicals. These unstable carbon-centered radicals can then rapidly interact with molecular oxygen to generate lipid peroxyl radicals, which ultimately extract hydrogen atoms from other molecules through Fenton reactions, resulting in the formation of lipid hydroperoxides[18]. Secondly, IR upregulates the expression of acyl-CoA synthetase long-chain family member 4 (ACSL4), promoting the biosynthesis of phospholipids containing polyunsaturated fatty acids, although the precise mechanisms underlying the IR-induced increase in ACSL4 levels remain unclear [14]. Lastly, IR also causes the depletion of glutathione (GSH), which undermines the iron death defense mediated by glutathione peroxidase 4 (GPX4), further facilitating ferroptosis [17]. In summary, multiple lines of evidence from various studies establish a robust link between ferroptosis and ionizing radiation.

Ferroptosis is an iron-dependent form of cell death that causes iron and lipid peroxidation toxicity through massive lipid peroxidation-mediated damage. As an evolutionarily conserved programme, ferroptosis plays a critical role in both development and disease in a variety of organisms [19,20]. It has been reported that Chinese medicines and their active ingredients that intervene in ferroptosis have the characteristics of regulating multiple targets, structural stability, high safety, low price and easy availability. By exploring the targets of Chinese medicines to intervene in diseases through the ferroptosis mechanism, it can provide a theoretical basis for research to explore new therapies for diseases [21].

Flavonoids are a group of naturally occurring substances found in plant foods such as fruits, vegetables and flowers that have iron chelating activity[22,23]. Luteolin is a compound that

belong to the flavonoid group, which can be extracted from flowers, herbs, vegetables and spices [24]. Studies have shown that luteolin has a powerful antioxidant capacity that effectively scavenging free radicals and protecting cells [25]. In a previous study, we screened the activity of total extracts of Ginkgo biloba flowers and their different parts and compounds using an Erastin induced ferroptosis model in vascular endothelial cells. We found that the total extract of the Ginkgo biloba flower as well as its chloroform site, ethyl acetate site and n-butanol fractions, exhibited anti-vascular endothelial cell ferroptosis activity. We found that luteolin has the most significant cytoprotective effect, inhibiting ferroptosis by regulating ferroptosis-related proteins [26]. Therefore, the present study hypothesizes that luteolin may have the potential to combat radiovascular injury.

## 2. Materials and methods

### 2.1. Cell culture and treatment

Purchase human umbilical vein endothelial cells (HUVECs) line from China Typical Culture Collection Center cell bank on the National Experimental Cell Resource Sharing Platform (Resource ID: 4201PAT-CCTCC00692). HUVECs were cultured in Dulbecco's Modified Eagle Medium (C11885500BT, Gibco). The culture medium was supplemented with 10% fetal bovine serum (FSP500, ExCell), 100 U/ml penicillin, and 100 μg/ml streptomycin (P1400, Solarbio).

Ferroptosis was induced in HUVECs by treatment with Erastin (S7242, selleck) at a concentration of 5μM. Luteolin and fer-1 (S7243, Selleck), both at concentrations of 5μM and 1μM, respectively, were used as protective agents. After 24 hours, samples were taken for further analysis.

### 2.2. Animal model

Male C57BL/6 mice (7–8 weeks old) were purchased from SiPeiFu (Beijing, China). All animal experiments were approved by the animal laboratory of the Experimental Animal Centre of the Military Medical Research Institute (ethic number IACUC-DWZX-2022-847). Irradiation was performed with $^{60}$Co γ-rays at the Military Medical Research Institute. C57BL/6 mice were shaved on the upper back and irradiated locally with 20 Gy. The irradiation group was given luteolin (10 mg/kg) by gavage 24 hours before irradiation and 2 hours before irradiation. The material was taken and photographed after 3 days. Anesthesia and/or analgesia were administered via intraperitoneal injection of 1% pentobarbital sodium (0.1mL/20g). Efforts to alleviate pain and distress included providing appropriate analgesia and anesthesia during surgical procedures, as well as ensuring adequate housing conditions for the animals. The euthanasia method utilized was cervical dislocation. The animals were weighed daily and the skin condition of the back was observed.

### 2.3. Immunohistochemistry staining

Tissues were fixed in formalin, dehydrated, embedded in paraffin, and sliced into 4μm sections. The sections were deparaffinized in xylene and hydrated with alcohol. Citrate buffer was used for antigen retrieval, and 3% $H_2O_2$ was used to eliminate endogenous peroxidase activity. Sections were blocked with BSA, incubated with primary antibody overnight at 4˚C, washed with Phosphate Buffered Saline (BL302A,bioshark), incubated with peroxidase-conjugated secondary antibody for 15 min at room temperature, stained with diaminobenzidine working solution, counterstained with hematoxylin, dehydrated with alcohol, and mounted.

## 2.4. Cell viability analysis

Use the Cell Counting Kit 8 (40203ES60, YEASEN) to assess cell viability according to the instructions provided in the manual. Briefly, plate the cells in a 96-well plate and pre-treat the cells. 10 μL of CCK-8 reagent is added to each well, and then the plates are incubated for 3 hours at 37°C in a 5% $CO_2$ cell culture incubator. After incubation, the absorbance is measured at 450 nm using a microplate reader (Multiskan MK3, Thermo Fisher Scientific), and the results are calculated.

## 2.5. Giemsa staining

After pre-treatment, cells were fixed with 4% paraformaldehyde (P1110, Solarbio) for 30 minutes, stained with Giemsa staining solution (G1010, Solarbio) at room temperature for 30 minutes, washed with PBS, observed under Leica DMIRB, and images captured.

## 2.6. MDA assay

The MDA assay kit (S0131, Beyotime) was used to measure lipid peroxidation levels according to the manufacturer's instructions. Briefly, $2 \times 10^6$ treated or transfected cells were collected and incubated with 150 μl lysis buffer on ice for 30 minutes. Subsequently, 100 μl of sample supernatant was combined with 200 μl of TBA working solution and incubated at 100°C for 15 minutes, while the remaining supernatant was used for protein concentration analysis. After centrifugation, the supernatant was collected, and the MDA content was determined by measuring the absorbance at 532 nm. The MDA concentration was calculated, and the relative MDA content was obtained by dividing the MDA concentration by the protein concentration.

## 2.7. Total Superoxide Dismutase Assay

Follow the instructions in the Total Superoxide Dismutase Assay Kit with WST-8 (S0101S, Beyotime). The cells are washed once with ice-cold PBS, followed by cell lysis with the superoxide dismutase (SOD) sample preparation solution. The resulting suspension is centrifuged at 12,000g for 5 minutes at 4°C, and the supernatant is collected after centrifugation. The pre-set detection reaction system is then prepared by adding the appropriate liquids to a 96-well plate. The plate is then incubated at 37°C for 30 minutes. After incubation, the absorbance is measured at 450 nm using a microplate reader. Finally, the total SOD activity is calculated from the absorbance values.

## 2.8. NADP$^+$/NADPH assay

Perform the experiment according to the procedure described in the NADP$^+$/NADPH Assay Kit with WST-8 (S0179, Beyotime) instruction manual. The old culture medium is aspirated, and the cells are lysed with NADP$^+$/NADPH extraction solution. The lysed suspension is then centrifuged at 12,000g for 10 minutes at 4°C. 50μL of the test sample is taken and added to a 96-well plate for the determination of the total NADP$^+$ and NADPH levels. Another 200μL of the test sample is taken and heated in a water bath at 60°C for 30 minutes. The pre-set detection reaction system is then added to the 96-well plate. The plate is incubated for 10 minutes at 37°C in the dark. 10μL of the color developer is added to each well, mixed well, and the plate is incubated in the dark at 37°C for 20 minutes. The absorbance is then measured at 450nm using a microplate reader. Finally, the NADP$^+$/NADPH levels are calculated from the absorbance values.

### 2.9. Lillie's ferrous iron stain

After pre-treatment, cells are fixed with 4% paraformaldehyde for 30 minutes. Subsequently, Lillie's Staining Solution (G3320, Solarbio) is then added to the well plate and incubated for 30 minutes at room temperature. The cells are washed three times with distilled water. Nuclear Fast Red is added to the well plate, which is then incubated for 10 minutes at room temperature. The cells are washed twice with distilled water. The well plate is then placed under the Leica DMIRB for observation and photography.

### 2.10. Dihydroethidium (DHE)

Pretreated HUVECs were exposed to dihydroethidium (S0063, Beyotime) at a concentration of 10 μM. The mixture was incubated at a temperature at 37˚C for 30 min. It was then rinsed twice with PBS and fixed with 4% paraformaldehyde fixative for a further 30 minutes. The coverslip was placed on the slide and sealed with an anti-fluorescence quenching agent containing DAPI. Laser scanning confocal microscopy (LSCM) was used for examination and image acquisition.

### 2.11. MitoTracker assay

The MitoTracker® Red CM-H2XRos mitochondrial red fluorescent probe assay (C1049B, Beyotime) was performed to examine the morphology and functional status of mitochondria. The HUVECs were pre-treated, and the working fluid was heated to 37˚C and added to the medium a final concentration of 500 nM. Incubation was performed at 37˚C in 5% $CO_2$ incubator for 45 min. The well plate was then transferred to a 37˚C, 5% $CO_2$ cell culture incubator and incubated for 45 minutes. Remove the culture medium, add 4% paraformaldehyde to the well plate, and incubate at 37˚C for 30 minutes. Transfer the coverslips to slide mounts and seal them with an anti-fade mounting medium containing DAPI. Observe and record images using LSCM.

### 2.12. Western blot

Following pre-treatment, HUVECs were solubilized in radioimmunoprecipitation assay buffer containing protease inhibitors for 30 min on ice. Equal amounts of proteins were separated by sodium dodecyl sulfate-polyacrylamide gel electrophoresis and transferred to polyvinylidene fluoride membranes, blocked with 5% skim milk solution, and incubated in primary antibodies overnight at 4˚C. The dilution ratio of the primary antibody is 1:1000. The membrane was incubated with a secondary antibody for 1 hour at room temperature. In addition, the dilution ratio of the secondary antibody was 1:10 000.ECL detection reagent was utilized to develop the bands. GAPDH (Glyceraldehyde-3-phosphate dehydrogenase, A19056, ABclonal) was used as the loading control to show that the loaded protein was similar across the gel in western blotting. The bands were quantitatively analyzed using ImageJ (1.4.3.67), and the gray value of the target protein was divided by the gray value of GAPDH for calculation and analysis.

### 2.13. Drug Affinity Responsive Target Stability (DARTS)

The old culture medium in the dish was discarded on ice. The cells were washed three times with ice-cold PBS. Lysis buffer was added to the dish, and the cell suspension collected with a cell scraper was transferred to a centrifuge tube for complete lysis on ice, with periodic vortexing of the cell suspension. After lysis, the cell suspension was centrifuged at 18,000 g for 10 minutes at 4˚C. The supernatant was transferred to a new Eppendorf tube. The supernatant was mixed with DMSO/luteolin in a 99:1 ratio. The mixture was placed on a rotator and

incubated at room temperature for 20 minutes. The 10× TNC was diluted to 1× TNC with ddH2O, and the protease was then diluted in 1× TNC. After sampling, the input group samples were directly mixed with 5× SDS-PAGE buffer and heated at 99˚C for 10 minutes; the remaining samples were diluted with the prepared protease according to the experimental design, ensuring that the digestion time was consistent. The samples were incubated at room temperature for 15 minutes before adding 5× SDS-PAGE buffer and heating at 99˚C for 10 minutes. The gel was prepared according to the instructions of the Omni-Easy™ One-Step PAGE Gel Quick Preparation Kit (PG222, Epizyme Biomedical). The electrophoresis buffer was prepared by mixing 900 mL of distilled water with 100 mL of electrophoresis buffer (PS105S, Epizyme Biomedical). Each sample must have protein markers on both sides, according to the sample order. The voltage for electrophoresis was set to 120 V. Electrophoresis was stopped when the marker had migrated approximately 1 cm. The target bands were cut into 1×1 cm pieces and placed into 1.5 mL Eppendorf tubes. A drop of distilled water was added to the gel-containing Eppendorf tubes, which were then labeled. The samples were subjected to protein mass spectrometry analysis (Beijing Proteome Research Center). The remaining protein samples were used for subsequent electrophoresis experiments for further validation.

## 2.14. Surface Plasmon Resonance (SPR)

The SPR detection instrument used is the Reichert 4SPR, with a dextran chip, operating at 25˚C with 2% DMSO in PBST as the running buffer. The protein immobilization conditions for HSPB1 are as follows: 40mg EDC mixed with 10mg NHS in 1mL of deionized water, injected at a flow rate of 10μL/min for 7min; the protein is diluted to 50μg/mL in 10mM sodium acetate (pH 5.0), injected at 10μL/min for 7min; blocked with 1M ethanolamine at pH 8.5 for 7min. For binding, luteolin is diluted in PBST containing 2% DMSO in a 2-fold gradient, injected at a flow rate of 25 μL/min, with a binding time of 1 min and a dissociation time of 2 min.

## 2.15. Gene Ontology (GO) and Kyoto Encyclopedia of Genes and Genomes (KEGG) functional enrichment analysis

Biological function exploration of target genes was conducted using the clusterProfiler package (version 3.14.3) through GO and KEGG functional enrichment analysis [27]. Gene ID conversion was performed using the org.Hs.eg.db package (version 3.10.0), and Z-scores for enrichment pathways were calculated using the GOplot package (version 1.0.2) to assess the significance of correlation [28]. Pathways were considered significantly enriched if they met the criteria of false discovery rate (FDR)<0.25 and adjusted p-value<0.05.

## 2.16. Data analysis

All data were checked for normal distribution, expressed as mean ± standard deviation (SD) and then analysed using Student's t-test or one-way ANOVA followed by post-hoc test. Correlational analyses were performed using Pearson product-moment correlation. All statistical analyses were performed using GraphPad Prism 8.0.2. Values $p < 0.05$ were considered statistically significant.

## 3. Results

### 3.1. Luteolin ameliorates radiovascular injury and promotes microvessels angiogenesis in mice

In our previous results, we found that luteolin has good antioxidant capacity. When IR interacts with living matter, a stress response can occur through, for example, the formation of

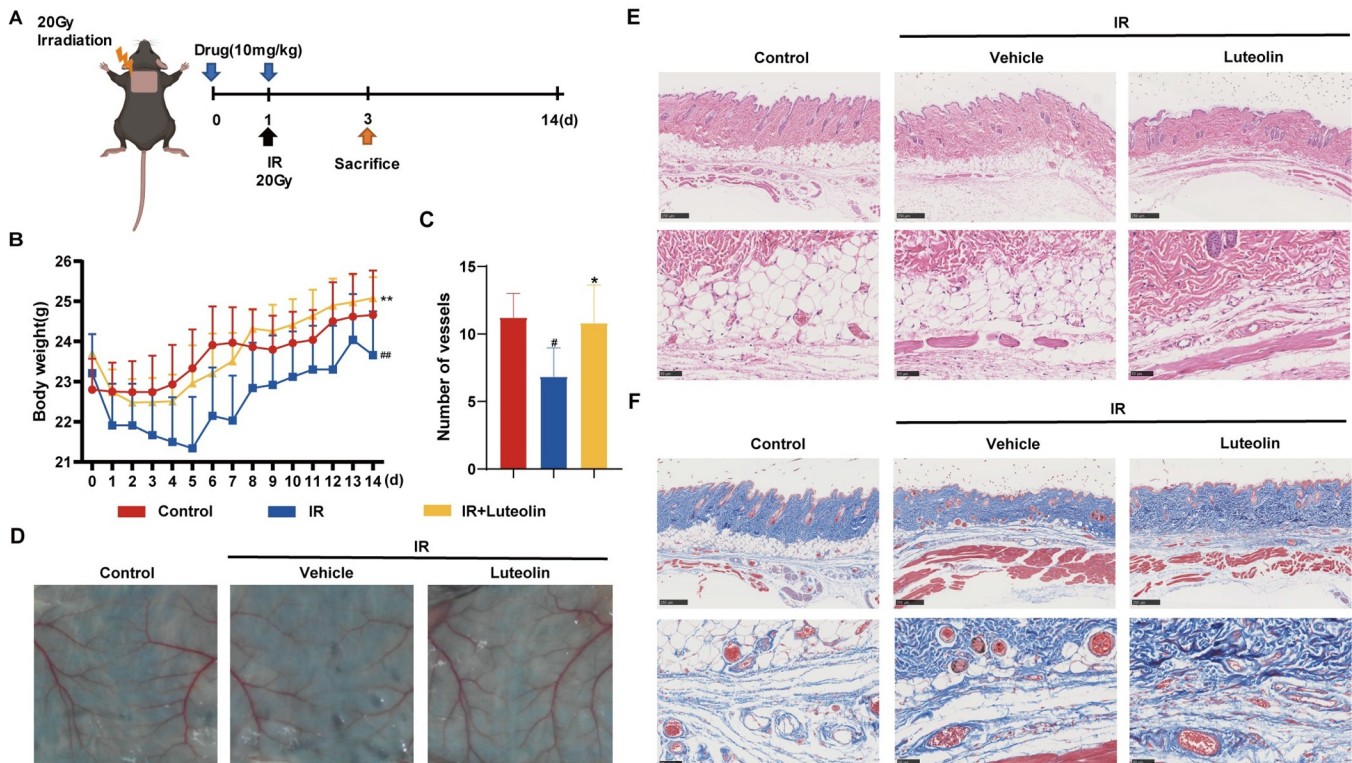

**Fig 1. Protective effects of luteolin on radiation vascular injury.** A. C57BL/6 mice were shaved on the upper back and locally irradiated with 20 Gy of $^{60}$Co γ-rays, and pre-treated mice were given luteolin by gavage (10 mg/kg, administered by gavage 24 hours before irradiation and 2 hours prior to irradiation for a total of 2 doses). B. Changes in body weight of mice after irradiation with 20 Gy $^{60}$Coγ rays. Representative graphs of vascular tissue 3 days after irradiation with or without Luteolin (C), H&E staining (E), and Masson trichrome staining (F). Scale bar = 250 μm, cropped image scale bar = 50 μm. #vs control, *vs Ionizing radiation (IR). (#$p$<0.05, ##$p$<0.01, *$p$<0.05, **$p$<0.01).

reactive oxygen species. Therefore, we speculate that luteolin may have a protective effect against radiovascular injury. Therefore, we referenced the modeling approach of Kim et al [29] to establish a mouse model of radiovascular injury by irradiating the skin of the upper back of mice with 20Gy $^{60}$Co gamma rays. Luteolin was administered by gavage 24h and 2h before irradiation (Fig 1A). Daily weighing of the mice during the period from 24 h before irradiation to the 14th day after irradiation revealed a decrease in body weight after irradiation, but after the administration of luteolin, the body weight of the mice was higher than that of the irradiated group and even exceeded that of the control group after the eighth day (Fig 1B and 1C). On the third day after irradiation, mice were dissected and sampled, and analysis of the skin tissue revealed a reduction in terminal microvessels after irradiation, whereas images of the luteolin group showed neovascularization at the ends of the vessels (Fig 1D). The degree of inflammation and collagen deposition in irradiated skin was analyzed by hematoxylin-eosin (H&E) and Masson's trichrome staining, which showed dilatation of the lumen, congestion of the lumen, swelling of vascular endothelial cells, reduction in the density of microvessels, hyperplasia of subcutaneous myofibers, and reduction in ganglia after irradiation. However, after administration of luteolin, microvascular neovascularisation, collagen fiber proliferation, vitrification and thinning of the fat layer were found (Fig 1E). The results of Masson trichrome staining showed disturbed collagen fiber arrangement under the skin and perivascular collagen deposition in the irradiated group, while luteolin had an ameliorating effect (Fig 1F). Vessel number statistics showed a significant reduction in the irradiated group, whereas the number

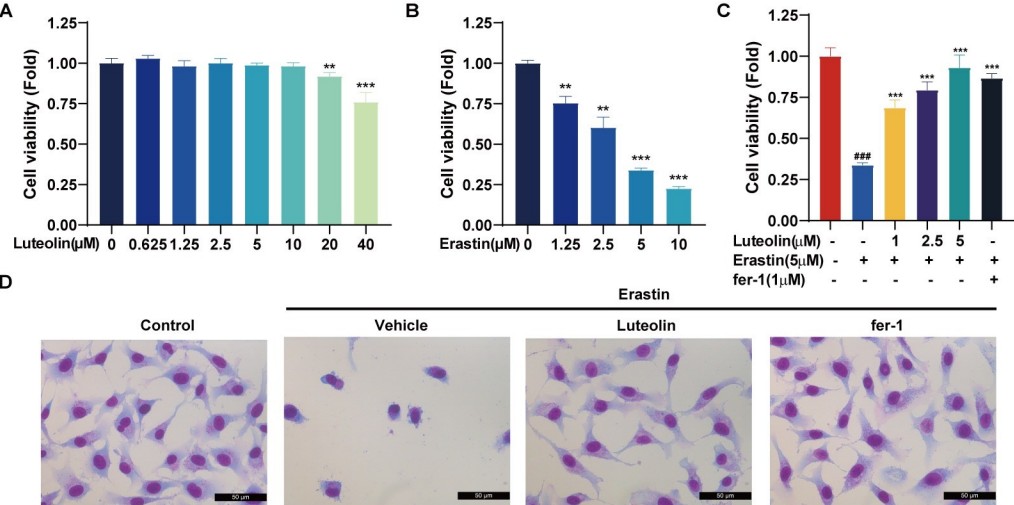

**Fig 2. Luteolin has anti-ferroptosis ability and low toxicity.** A. Cytotoxicity of human umbilical vein endothelial cells (HUVEC) after treatment with different concentrations of luteolin was detected by Cell Counting Kit 8 (CCK8) assay, n = 5, * vs control. B. The effect of cell viability of HUVEC after treatment with different concentrations of Erastin was detected using the CCK8 assay, n = 5, * vs control. C. Effect of Luteolin and Ferrostatin-1 (fer-1) on cell survival in Erastin-induced ferroptosis in HUVEC by CCK8 assay, n = 5, * vs Erastin. D. Cellular morphological changes were observed using the Giemsa staining method, scale bar = 50 μm. (**$p<0.01$, ***$p<0.001$, ### $p<0.001$).

of vessels after luteolin administration was not significantly different from the control group (Fig 1D). In summary, the experimental results showed that luteolin was able to ameliorate radial vascular injury and promote microvascular neovascularisation and collagen fiber proliferation in mice.

### 3.2. Luteolin can resist ferroptosis

We have shown that luteolin has a protective effect against radiovascular injury in vivo, but it is not clear how luteolin works. Therefore, the mechanism of action of luteolin was investigated in vitro by studying its protective effect on HUVECs. First, toxicity assays were performed and the results showed low toxicity of luteolin to HUVEC at different concentrations (Fig 2A). Further studies revealed that Erastin induced ferroptosis in HUVEC at an IC50 of 5 μM (Fig 2B), whereas luteolin at 5 μM exhibited superior anti-ferroptosis ability at a concentration, even surpassing the ferroptosis inhibitor ferrostatin-1 (fer-1) (Fig 2C). Therefore, in subsequent experiments of this study, the concentration of luteolin was set at 5 μM, the concentration of Erastin at 5 μM, and the concentration of fer-1 at 1 μM. Morphological observations showed that luteolin and fer-1 were effective in reversing the manifestations of Erastin-induced cellular ferroptosis, including nuclear and cytoplasmic sequestration and complete loss of cytoplasmic structures. These findings reveal that luteolin protects cells from ferroptosis damage.

### 3.3. Luteolin has a strong antioxidant damage ability

The main features of ferroptosis include lipid oxidation and accumulation of divalent iron ions. The present study was conducted to investigate the role of luteolin in the process of ferroptosis. The experimental results showed that luteolin could effectively reverse Erastin-induced lipid oxidation in HUVECs and reduce MDA levels (Fig 3A). Further studies revealed that luteolin was able to restore SOD activity in HUVECs and reduce the reduction of SOD

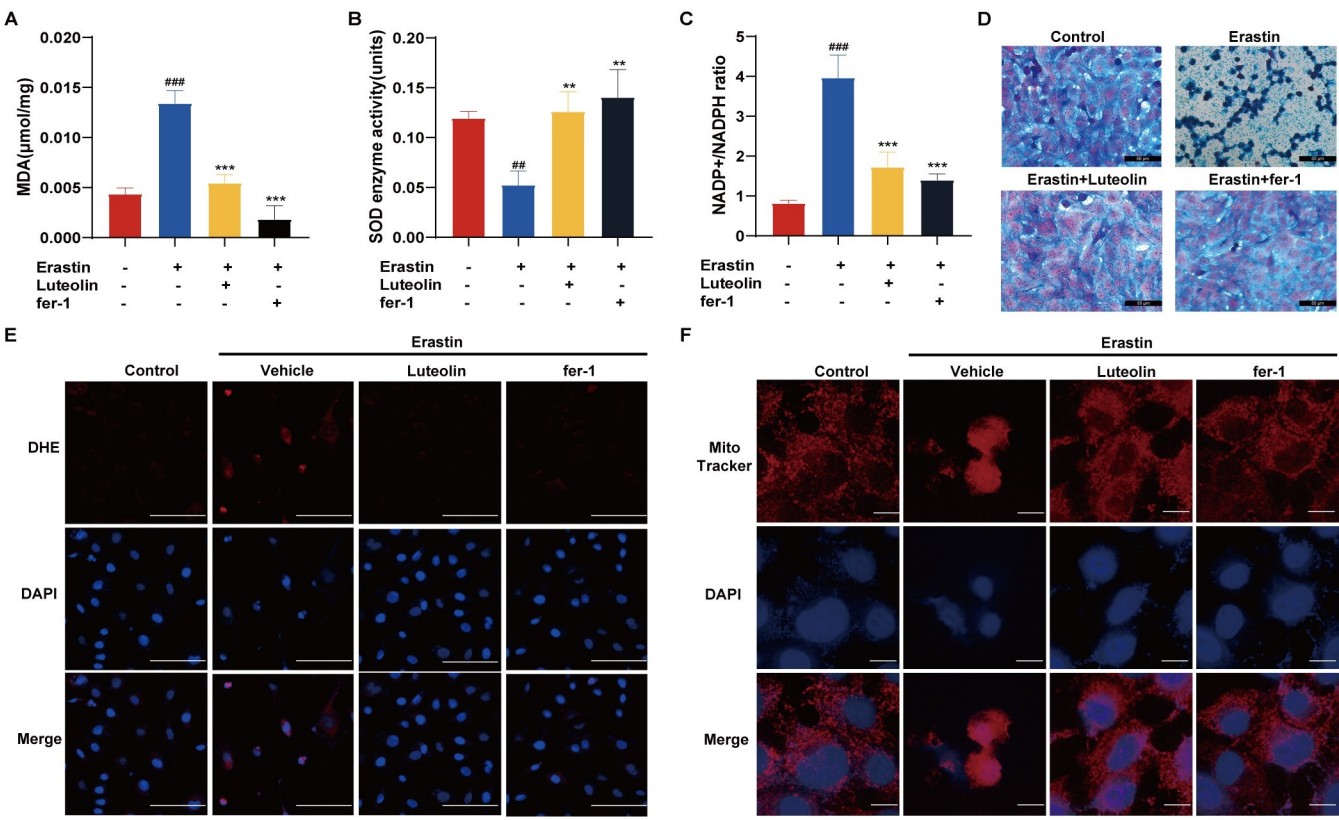

**Fig 3. Luteolin can resist oxidative damage.** A. MDA detection of lipid oxidation after luteolin rescue of HUVECs undergoing ferroptosis, n = 3. B. Changes in SOD enzyme activity after luteolin rescue of HUVECs from ferroptosis, n = 3. C. NADP$^+$/NADPH ratio detects changes in intracellular redox state following luteolin rescue of HUVECs from ferroptosis, n = 3. D. Fe$^{2+}$ staining to detect intracellular divalent iron accumulation after luteolin rescue of HUVEC from ferroptosis. E. DHE assay of luteolin rescues the level of oxidative stress in cells after ferroptosis in HUVECs. Scale bar = 100 μm. F. MitoTracker to observe the morphology of mitochondria after luteolin rescue of HUVECs from ferroptosis. Scale bar = 10 μm. #vs control, *vs Erastin. (##$p<0.01$, ###$p<0.001$, **$p<0.01$, ***$p<0.001$).

caused by Erastin (Fig 3B). Fluorescence microscopy observation showed that the addition of luteolin attenuated the DHE fluorescence intensity during ferroptosis, indicating that luteolin had a significant effect on inhibiting lipid oxidation (Fig 3E). In addition, the NADP$^+$/ NADPH ratio was reduced after luteolin intervention, suggesting that luteolin can regulate the intracellular redox state (Fig 3C). In addition, the experimental results also showed that the addition of luteolin was effective in reversing the accumulation of divalent iron ions during ferroptosis (Fig 3D). Furthermore, ferroptosis is accompanied by mitochondrial consolidation, which is manifested by mitochondrial shrinkage and compact structure. In the present study, the morphological structure of intracytoplasmic mitochondria was improved by the addition of luteolin (Fig 3F). In conclusion, luteolin has significant antioxidant capacity and can effectively reverse cellular damage during ferroptosis.

### 3.4. Identification of luteolin target proteins by the DARTS method

DARTS is a new target discovery method that reveals drug target interactions in cells or tissues by tracking changes in the stability of proteins that act as receptors for biologically active small molecules [30]. To investigate which proteins luteolin luteolin targets for its antioxidant effects, we used the DARTS technique to search for potential targets of luteolin. After we lysed the cells to collect the proteins, we allowed the proteins to bind to DMSO and luteolin, respectively, and

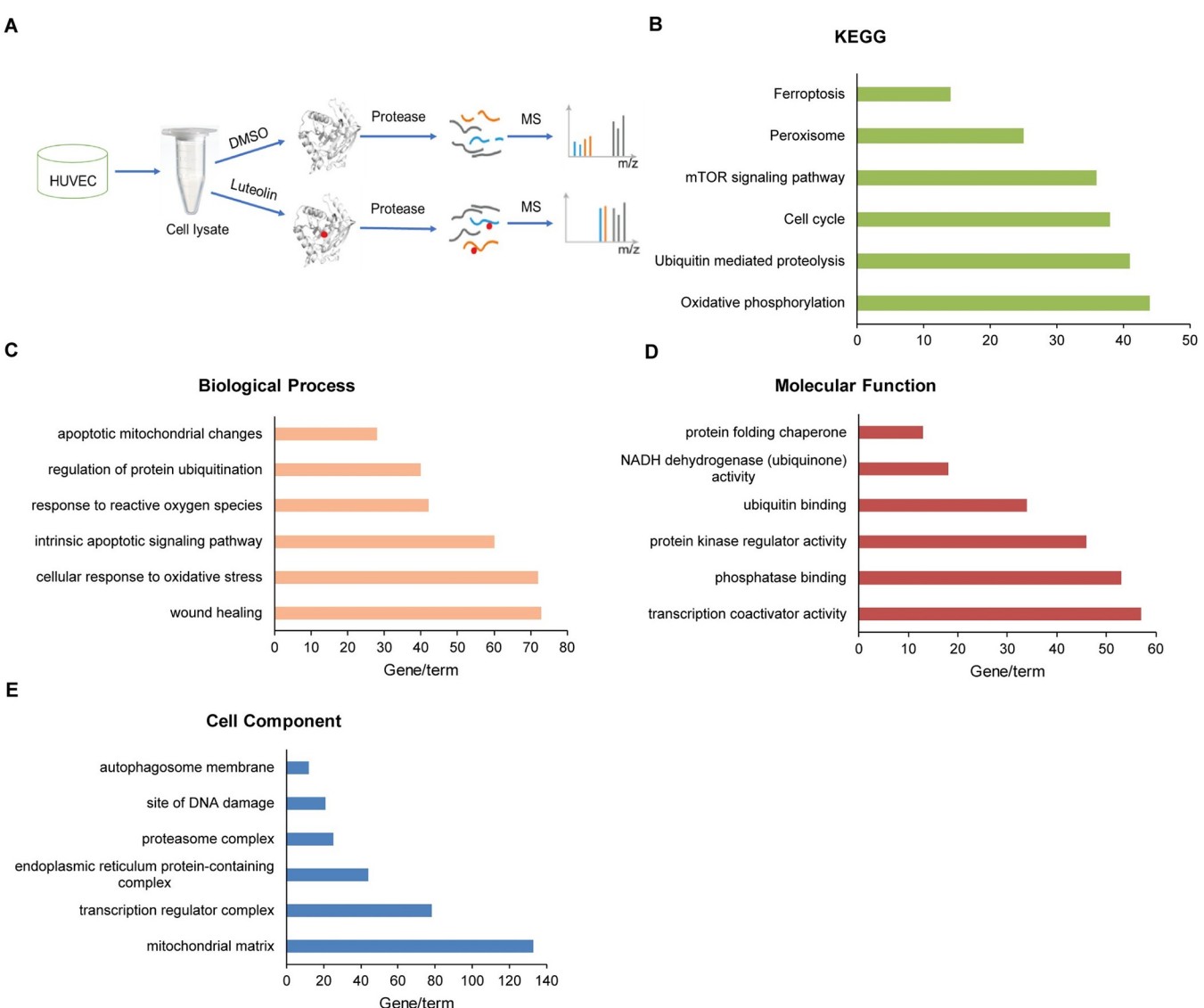

**Fig 4. Drug affinity responsive target stability (DARTS) identification of target acting proteins of luteolin.** A. Schematic representation of the DARTS process, with a total of 5874 luteolin potential target proteins identified. B-E. Protein enrichment terms after GO, and KEGG pathway analysis are shown on DAVID.

when the target proteins bind to luteolin, luteolin protects the target proteins from being hydrolyzed by proteases (Fig 4A). We identified 5874 potential target proteins by mass spectrometry (S3 File). We then performed GO and KEGG pathway analyses of these candidate target proteins using DAVID and identified six terms regulated by luteolin in Molecular Function, Biological Process, Cell Component, and KEGG pathway, as shown in Fig 4B–4E. These results suggest that potential luteolin target proteins are closely related to mitochondrial function.

### 3.5. HSPB1 is one of the target proteins of luteolin and is involved in the regulation of ferroptosis

Based on coverage, peptide and abundance screening in the DARTS results, it was found that heat shock protein beta-1 (HSPB1) was found to be one of the target proteins of luteolin.

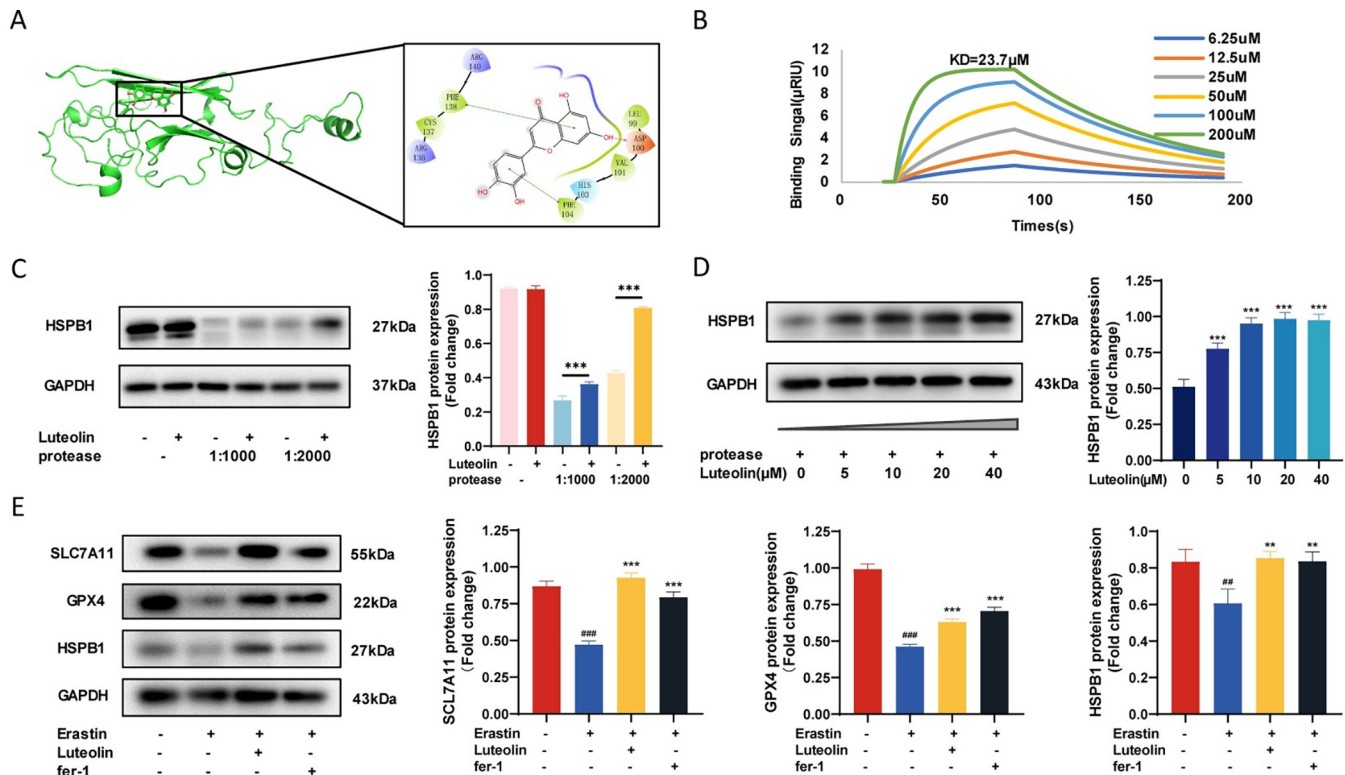

**Fig 5. HSPB1 is one of the target proteins of luteolin.** A. Schematic illustration of how computer simulated docking luteolin binds to HSPB1. B. Fitted curves of the interaction results of luteolin with HSPB1 detected by SPR technique. C. Western Blot validation of DARTS results, right panel shows grey value analysis, *vs control. D. Western Blot assay for luteolin affinity for HSPB1, grey value analysis on the right, *vs control. E. Western Blot detection of luteolin-regulated ferroptosis-associated proteins. #vs control, *vs Erastin. (##$p<0.01$, ###$p<0.001$, **$p<0.01$, ***$p<0.001$).

HSPB1 is a member of the small heat shock protein family and plays a protective role in cellular stress situations, e.g. as an antioxidant and anti-apoptotic agent in oxidative and chemical stress, respectively [31,32]. Computer simulation docking results by computer simulation showed that HSPB1 and luteolin bind through mediator stacking and hydrogen bonding (Fig 5A). The interaction between luteolin and HSPB1 protein was detected by the surface plasmon resonance (SPR) technique, and the results showed that they were bound with an affinity of 23.7 μM (Fig 5B). Western blot results experiments showed that the addition of luteolin under the same protease concentration conditions reduced the hydrolysis of HSPB1, consistent with the DARTS results (Fig 5C). The expression of HSPB1 increased with increasing luteolin concentration when different concentrations of luteolin were added at a protease concentration of 1:1000 (Fig 5D). It was also found that the expression of SLC7A11, GPX4, and HSPB1 was significantly decreased after the addition of Erastin, and the expression of SLC7A11, GPX4, and HSPB1 was elevated after the addition of luteolin and fer-1, which inhibited ferroptosis (Fig 5E). The above results indicate that luteolin has an affinity for HSPB1, which is one of the target proteins of luteolin and is involved in the regulation of ferroptosis.

## 4. Discussion

Antiradiation drugs are designed to reduce the damage caused by radiation to living organisms. Currently, amifostine is the only antiradiation drug approved for use by the US Food and Drug Administration, but its clinical use is limited due to its serious adverse effects

[12,33]. Antiradiation drugs can be classified, depending on their origin, as traditional and natural medicines, chemical drugs and biological products[12]. Studies have shown that flavonoids have good radioprotective and sensitizing effects [34]. One of the active ingredients is luteolin. Luteolin has been used in the treatment of a wide range of diseases, including various human malignancies such as lung and breast cancer, as well as skin aging, skin cancer and inflammatory skin diseases [35,36]. In this study, we used the upper back of irradiated mice to establish a model of radioactive vascular injury. We found that luteolin could promote microvascular neovascularisation, prevent radioactive vascular injury, and repair radiation-induced vascular injury and restore microvascular density to normal levels after 3 days of radiation treatment, suggesting that luteolin could alleviate the symptoms of radioactive vascular injury and has potential value in the field of radiation.

The main threat to cells from irradiation is reactive oxygen species, which interact with biomolecules to produce secondary free radicals that cause oxidative damage to RNA, DNA, proteins and lipids [37]. Irradiation effectively induces intracellular lipid peroxidation and ferroptosis, suggesting that ferroptosis is at least as important as other forms of cell death induced by irradiation [14]. Therefore, in the present study, we sought to investigate whether luteolin protects against radiogenic injury by inhibiting ferroptosis. Cell viability and morphological changes showed that luteolin reduced Erastin-induced ferroptosis in HUVE *in vitro*. In addition, luteolin elevates MDA levels and $NADP^+$/NADPH ratio. Moreover, luteolin was able to reduce SOD enzyme activity, decrease divalent iron accumulation, reduce DHE levels and improve mitochondrial morphological changes. These results suggest that luteolin may attenuate ferroptosis by inhibiting ROS generation, reducing lipid peroxidation, modulating iron metabolism, maintaining mitochondrial morphology and reducing cell death.

Although the anti-peroxidation ferroptosis activity of luteolin has been confirmed, its specific mechanism of action remains unknown. To address this issue, this study used DARTS technology, a labeling-free small molecule probe technology that tracks and identifies changes in the stability of target proteins upon binding of small molecules. DARTS technology has the advantages of simplicity, efficiency, versatility and accuracy [30]. With this technique, we found that HSPB1 is one of the targets of luteolin's action. In addition, computer simulations of docking and Western Blot results further confirmed this finding. Yuan et al [38] proposed that HSPB1 upregulation inhibited Erastin-induced ferroptosis in glioblastoma cells, whereas HSPB1 knockdown had an enhancing effect. It has also been shown that HSPB1 overexpression reduced iron content in hippocampal tissues of hypoxic-ischemic rats, as well as in hippocampal neurons mediated by oxygen-glucose deprivation *in vitro* [39]. Liang et al. [32] It demonstrated that HSPB1 overexpression inhibits ferroptosis, which ultimately leads to reduced sensitivity of breast cancer cells to doxorubicin. The above studies suggest that HSPB1 may be a negative regulator of ferroptosis. Previous studies have shown that protein kinase C-mediated phosphorylation of HSPB1 protects against ferroptosis by reducing iron-mediated production of lipid reactive oxygen species [31]. HSPB1 also binds to Ikβ-α and promotes its ubiquitination-mediated degradation, leading to increased nuclear translocation and activation of the NF-κB signaling pathway, with novel functional roles in regulating chemoresistance and ferroptosis in breast cancer [32]. This was confirmed by our findings that Erastin-mediated down-regulation of HSPB1 in ferroptosis was elevated by the addition of luteolin. Thus, luteolin may inhibit ferroptosis by targeting HSPB1. SLC7A11 maintains the production of GSH, a major endogenous antioxidant, through a series of reactions following the exchange of extracellular cystine for intracellular glutamate. Inhibition of SLC7A11 expression by the small molecule compound Erastin leads to GSH depletion, which triggers ferroptosis [40]. Inhibition of the SLC7A11 pathway is one of the most critical upstream mechanisms for the induction of ferroptosis. GPX4 is a selenoprotein, and since GPX4 is the major phospholipid hydroperoxide

(PLOOH)-neutralizing enzyme, Erastin indirectly inactivates GPX4 by inhibiting cystine uptake, thereby depriving cells of cysteine. Thus, accumulation of PLOOHs may lead to rapid and irreparable damage to the plasma membrane, resulting in cell death [41,42]. Here, luteolin inhibited ferroptosis by suppressing the reduced expression of SLC7A11 and GPX4 induced by Erastin.

Taken together, our results suggest a potential mechanism by which luteolin exerts antioxidant damage ferroptosis by targeting the HSPB1-regulated SLC7A11/GPX4 axial pathway, but its fine regulatory mechanism remains to be further explored. Meanwhile, the present study found that the pathway enriched in potential target proteins of luteolin, indicating that luteolin may also exert its antioxidant stress damage effect by regulating deubiquitinating enzymes to stable target proteins, which will be further explored in the future.

In summary, our results indicate that luteolin exerts its potential mechanism against oxidative damage and ferroptosis by targeting the HSPB1, which regulates the SLC7A11/GPX4 axis. However, the intricate regulatory mechanisms remain to be further elucidated. Additionally, this study revealed that the pathways enriched with potential target proteins of luteolin suggest it may also exert its antioxidative effects by modulating deubiquitinases to stabilize target proteins, warranting further investigation.

Future research could delve deeper into the regulatory mechanisms of luteolin on the HSPB1 and SLC7A11/GPX4 signaling pathways, as well as its precise role in the ferroptosis process. Such insights would enhance our understanding of luteolin's mode of action. Beyond the known targets, future studies might explore other potential targets associated with luteolin and its comprehensive mechanisms in radiation damage and ferroptosis. This would contribute to a holistic understanding of luteolin's biological effects.

Furthermore, the radioprotective capability of luteolin presents new perspectives and potential therapeutic strategies in the broader context of radiation damage treatment. Specifically, future research could focus on combination therapy strategies involving luteolin and other antioxidants or drugs to identify more effective treatment regimens for radiation injuries. Based on the findings, it is also recommended to conduct clinical trials to validate the practical application potential of luteolin in radiation damage therapy. This would provide additional scientific support for developing more effective strategies for treating radiation injuries and advance the exploration of luteolin in clinical applications.

## 5. Conclusion

Our study provides preliminary evidence for the protective effect of luteolin against vascular oxidative damage and elucidates the mechanism of action and potential targets of luteolin against ferroptosis. It provides a theoretical basis for the expanded application of luteolin as a small molecule candidate for the therapy of radiation-induced oxidative damage diseases.

## Supporting information

**S1 Graphical abstract.**
(TIF)

**S1 File. Western blot of raw data.**
(PDF)

**S2 File. Bar chart raw data.**
(XLSX)

**S3 File. Protein analysis.**
(XLSX)

## Author Contributions

**Conceptualization:** Lisheng Wang, Fengjun Xiao.

**Data curation:** Weiyuan Zhang, Yiming Wang.

**Formal analysis:** Li Wen, Changchang Lv.

**Funding acquisition:** Fengjun Xiao.

**Methodology:** Li Wen.

**Project administration:** Lisheng Wang.

**Resources:** Min Li.

**Software:** Weiyuan Zhang.

**Supervision:** Min Li.

**Validation:** Jia Hu, Tao Chen.

**Visualization:** Lisheng Wang.

**Writing – original draft:** Li Wen, Jia Hu, Yiming Wang, Changchang Lv.

**Writing – review & editing:** Min Li, Lisheng Wang, Fengjun Xiao.

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
