## [Decision Letter · Decision Letter 0]

6 Aug 2024

PONE-D-24-25996Luteolin target HSPB1 regulates endothelial cell ferroptosis to protect against radiation vascular injuryPLOS ONE

Dear Dr. Xiao,

Thank you for submitting your manuscript to PLOS ONE. After careful consideration, we feel that it has merit but does not fully meet PLOS ONE’s publication criteria as it currently stands. Therefore, we invite you to submit a revised version of the manuscript that addresses the points raised during the review process.

We look forward to receiving your revised manuscript.

Kind regards,

Jian Hao

Academic Editor

PLOS ONE

Journal Requirements:

This research was funded by the National Natural Science Foundation of China, grant number 82073489.

Reviewers' comments:

Reviewer's Responses to Questions

**Comments to the Author**

1. Is the manuscript technically sound, and do the data support the conclusions?

Reviewer #1: Partly

Reviewer #2: Yes

2. Has the statistical analysis been performed appropriately and rigorously? 

Reviewer #1: Yes

Reviewer #2: No

3. Have the authors made all data underlying the findings in their manuscript fully available?

Reviewer #1: Yes

Reviewer #2: Yes

4. Is the manuscript presented in an intelligible fashion and written in standard English?

Reviewer #1: No

Reviewer #2: No

5. Review Comments to the Author

**Reviewer #1: **Based on the previous work, the authors of this manuscript studied the detail mechanism of luteolin in protecting radiation vascular injury through targeting HSPB1-reuglated SLC7A11/GPX4 axial pathway. Basically, after proving that luteolin ameliorates radiation-induced vascular injury and promotes micro angiogenesis, the authors focused on the association between luteolin and ferroptosis. The key experiment is DARTS as the authors identified the key proteins targeted by luteolin. To the end, the western blot results further proved the authors hypothesis. The workload of the study is enough. However, the writing of the manuscript is not good enough to be accepted by the journal Plos One now. The authors are suggested to revise the manuscript according to the concerns as follow:

Minor concerns:

1) In the Introduction section, the first sentence need reference to suooprt.

2) When introducing the formation of reactive oxygen interactions in the 1st paragraph of the Introduction section, Prof. Wang’s work (10.1016/j.jes.2020.09.022) is recommended to be cited.

3) When introducing the function of flavonoids, Prof. Li’s work studying the pharmacokinetics of flavonoids (10.1002/bmc.5488) should be cited.

4) In the last paragraph of the Introduction section, the authors said that they have reported the total extract of the Ginkgo biloba flower exhibited anti-vascular endothelial cell ferroptosis activity. The authors should cite the detail reference instead of just mentioning the previous work.

5) In section 2.1, the authors should revise the writing of the paragraph instead of just providing the names and the origins. Besides, the full name of PBS should be given, and the “4%” reads strange.

6) In section 2.2, the full named of DMEM should be given.

7) In section 2.5, the last two sentences should be revised as these two sentences contain none subject. Same to the last sentence of section 2.12.

8) In the Discussion section, line 349, the number “26” should be deleted.

Major concerns:

1) Detail experimental information should be given for the section 2.14 as the DARTS experiment reads the key experiment for this study. Besides, the authors used GO and KEGG analysis, the detail experimental information should also be provided.

2) The authors should give a graphic description to illustrate the important findings of the work.

**Reviewer #2: **Thank you for the opportunity to review the research manuscript entitled “Luteolin targets HSPB1 to regulate endothelial cell ferroptosis and protect against radiation vascular injury.” However, I have a few revisions that the authors should address.

2 Introduction: To enhance the flow between different topics, please ensure a smooth transition from the general discussion of radiation to the specific topic of ferroptosis.

3 Please indicate the usage concentration of each drug clearly, as well as the number of replicates conducted for each experiment.

4 Highlight the importance of these findings in relation to the broader field of radiation injury treatments. Furthermore, propose specific practical applications or outline potential next steps that could be taken based on the study's results.

5 Please ensure the correct use of punctuation, such as spaces. There are many instances of incorrect spacing throughout the article. Ensure consistent formatting and terminology throughout the document, particularly in lines 224 and 225.

6 Attention should be paid to language clarity to make sentences easier to understand. For example, “We found that the total extract of the Ginkgo biloba flower as well as its chloroform site, ethyl acetate site and n-butanol fractions, exhibited anti-vascular endothelial cell ferroptosis activity”. Additionally, modify line 192 for clarity (Membranes were developed with the developer).

6. PLOS authors have the option to publish the peer review history of their article (what does this mean?). If published, this will include your full peer review and any attached files.

Reviewer #1: No

Reviewer #2: No

---

## [Author Response · Author response to Decision Letter 0]

20 Sep 2024

PONE-D-24-25996

PLOS ONE

Dear Editor,

Thank you very much for providing us the opportunity to revise our manuscript entitled “Luteolin target HSPB1 regulates endothelial cell ferroptosis to protect against radiation vascular injury". We are grateful to you and reviewers for their careful review and valuable comments, which are very constructive and helpful for improving our manuscript. We have done the revision of the document according to the suggestions from the reviewers. The responses to the reviewer’s comments are listed below, and all the revision had been highlighted with color for inspection. We hope that with this response and the accompanying changes to the manuscript, that it will now be considered acceptable for publication in PLOS ONE.

Thanks again for your time, effort and professional comments on our manuscript.

Fengjun Xiao, Professor. 

Beijing Institute of Radiation Medicine,

The Affiliated Hospital of Qingdao University, 

Beijing, 100850, PR China 

To Editor:

Responses: We have modified it according to the requirements of PLOS ONE.

Responses: "The funders had no role in study design, data collection and analysis, decision to publish, or preparation of the manuscript." 

3. We note that your Data Availability Statement is currently as follows: All relevant data are within the manuscript and its Supporting Information files.Please confirm at this time whether or not your submission contains all raw data required to replicate the results of your study. Authors must share the “minimal data set” for their submission. PLOS defines the minimal data set to consist of the data required to replicate all study findings reported in the article, as well as related metadata and methods. 

Responses: We have provided the original data, including the unanalyzed pictures, tables, etc. 

4. Please ensure that you have an ORCID iD and that it is validated in Editorial Manager. To do this, go to ‘Update my Information’ (in the upper left-hand corner of the main menu), and click on the Fetch/Validate link next to the ORCID field. This will take you to the ORCID site and allow you to create a new iD or authenticate a pre-existing iD in Editorial Manager. 

Responses: We have created ORCID as required. 

5. PLOS ONE now requires that authors provide the original uncropped and unadjusted images underlying all blot or gel results reported in a submission’s figures or Supporting Information files.

Responses: We have provided the original uncropped and unadjusted images of all western blot results reports as requested.

To reviewer 1

1) In the Introduction section, the first sentence need reference to suooprt.

Responses: We sincerely appreciate the valuable comments. We have added references, see References 1-4.

2) When introducing the formation of reactive oxygen interactions in the 1st paragraph of the Introduction section, Prof. Wang’s work (10.1016/j.jes.2020.09.022) is recommended to be cited.

Responses: We have cited Professor Wang's work, check References 9.

3) When introducing the function of flavonoids, Prof. Li’s work studying the pharmacokinetics of flavonoids (10.1002/bmc.5488) should be cited.

Responses: We have referenced Professor Li's research, refer to Reference 23.

4) In the last paragraph of the Introduction section, the authors said that they have reported the total extract of the Ginkgo biloba flower exhibited anti-vascular endothelial cell ferroptosis activity. The authors should cite the detail reference instead of just mentioning the previous work.

Responses: We have made the necessary modifications and adjusted the position of references. Please consult lines 103-105 and Reference 26 for details.

5) In section 2.1, the authors should revise the writing of the paragraph instead of just providing the names and the origins. Besides, the full name of PBS should be given, and the “4%” reads strange.

Responses: We have revised the writing of the paragraph and added the brand number of the reagent to the paragraph of the corresponding experimental method. And we have written the full name of PBS.

6) In section 2.2, the full named of DMEM should be given.

Responses: We have given the full name of DMEM, see lines 111 through 112.

7) In section 2.5, the last two sentences should be revised as these two sentences contain none subject. Same to the last sentence of section 2.12.

Responses: Thank you very much for your thoughtful suggestions. We have made revisions to the last sentences of Section 2.5 (now Section 2.4) and Section 2.12 (now Section 2.11), as reflected in lines 143-146 and line 205.

8) In the Discussion section, line 349, the number “26” should be deleted.\\

Responses: We sincerely appreciate your reminder, and we have eliminated the number "26".

Major concerns:

1) Detail experimental information should be given for the section 2.14 as the DARTS experiment reads the key experiment for this study. Besides, the authors used GO and KEGG analysis, the detail experimental information should also be provided.

Responses: We have given detailed experimental information for the DARTS and GO and KEGG analysis, see sections 2.13 and 2.15.

2) The authors should give a graphic description to illustrate the important findings of the work.

Responses: We have drawn a graph depicting the important findings of this study, see figure Abstract.

To reviewer 2

2 Introduction: To enhance the flow between different topics, please ensure a smooth transition from the general discussion of radiation to the specific topic of ferroptosis.

Responses: We believe this is an excellent suggestion. We have added this section according to the Reviewer's recommendation. We have successfully transitioned from a general discussion of radiation to the specific topic of ferroptosis in the second paragraph of the introduction, as reflected in lines 59-82.

Research indicates that ferroptosis plays a critical role in radiation-induced cell death responses. Ionizing radiation triggers tumor cells to generate lipid reactive oxygen species (ROS), leading to the accumulation of lipid peroxides, which ultimately induces ferroptosis. This process is facilitated by the production of hydroxyl radicals from radiation exposure, which promotes lipid peroxidation. Additionally, inhibiting SLC7A11 or GPX4 using ferroptosis inducers can enhance the sensitivity of tumor cells to radiotherapy (RT).

Studies have demonstrated that RT significantly increases markers of lipid peroxidation, such as C11-BODIPY staining and malondialdehyde (MDA) levels in both cancer cells and tumor samples. After irradiation, cells exhibit morphological characteristics consistent with ferroptosis, including mitochondrial shrinkage and increased membrane density. Moreover, the application of ferroptosis inhibitors like ferrostatin-1 (fer-1) or deferoxamine (DFO) has been shown to partially restore the clonogenic survival of various cancer cell lines following RT.

Mechanistically, the induction of lipid peroxidation and ferroptosis by ionizing radiation occurs through three main pathways: lipid peroxidation, upregulation of acyl-CoA synthetase long-chain family member 4 (ACSL4), and depletion of glutathione (GSH). These pathways collectively contribute to the robust relationship between ferroptosis and ionizing radiation, as evidenced by substantial findings in multiple studies.

3 Please indicate the usage concentration of each drug clearly, as well as the number of replicates conducted for each experiment.

Responses: We have provided the concentration details in lines 301-302, and the number of experimental repetitions has been added to the figure caption.

4 Highlight the importance of these findings in relation to the broader field of radiation injury treatments. Furthermore, propose specific practical applications or outline potential next steps that could be taken based on the study's results.

Responses: Thank you for your constructive comments. We have emphasized the significance of the findings of this study in the field of radiation damage treatment in the last three paragraphs of the discussion. Additionally, the results propose practical applications and next steps.

In summary, our results suggest that luteolin may protect against oxidative damage and ferroptosis by targeting HSPB1, which regulates the SLC7A11/GPX4 axis, though the detailed mechanisms require further investigation. The study also indicates that luteolin might enhance its antioxidant effects by modulating deubiquitinases to stabilize target proteins.

Future research should explore luteolin's regulatory mechanisms on the HSPB1 and SLC7A11/GPX4 pathways and its role in ferroptosis, along with other potential targets related to luteolin’s effects on radiation damage. 

Additionally, luteolin’s radioprotective properties could lead to new therapeutic strategies in radiation damage treatment. Future studies could focus on combination therapies with luteolin and other agents for improved treatment regimens. Clinical trials are recommended to assess the practical applications of luteolin in radiation therapy, supporting the development of more effective treatment strategies.

5 Please ensure the correct use of punctuation, such as spaces. There are many instances of incorrect spacing throughout the article. Ensure consistent formatting and terminology throughout the document, particularly in lines 224 and 225.

Responses: We were really sorry for our careless mistikes. Thank you for your reminder. We have corrected the use of whitespace in the manuscript and made corrections to the formatting and terminology.

6 Attention should be paid to language clarity to make sentences easier to understand. For example, “We found that the total extract of the Ginkgo biloba flower as well as its chloroform site, ethyl acetate site and n-butanol fractions, exhibited anti-vascular endothelial cell ferroptosis activity”. Additionally, modify line 192 for clarity (Membranes were developed with the developer).

Responses: We have improved the clarity of the language to make it easier for the reader to understand. For details, see lines 101-103 and 213-214.

---

## [Decision Letter · Decision Letter 1]

27 Sep 2024

Luteolin target HSPB1 regulates endothelial cell ferroptosis to protect against radiation vascular injury

PONE-D-24-25996R1

Dear Dr. Xiao,

We’re pleased to inform you that your manuscript has been judged scientifically suitable for publication and will be formally accepted for publication once it meets all outstanding technical requirements.

Kind regards,

Jian Hao

Academic Editor

PLOS ONE

Additional Editor Comments (optional):

Reviewers' comments:

Reviewer's Responses to Questions

**Comments to the Author**

1. If the authors have adequately addressed your comments raised in a previous round of review and you feel that this manuscript is now acceptable for publication, you may indicate that here to bypass the “Comments to the Author” section, enter your conflict of interest statement in the “Confidential to Editor” section, and submit your "Accept" recommendation.

Reviewer #1: All comments have been addressed

Reviewer #2: All comments have been addressed

2. Is the manuscript technically sound, and do the data support the conclusions?

Reviewer #1: Yes

Reviewer #2: Yes

3. Has the statistical analysis been performed appropriately and rigorously? 

Reviewer #1: Yes

Reviewer #2: Yes

4. Have the authors made all data underlying the findings in their manuscript fully available?

Reviewer #1: Yes

Reviewer #2: Yes

5. Is the manuscript presented in an intelligible fashion and written in standard English?

Reviewer #1: Yes

Reviewer #2: Yes

6. Review Comments to the Author

Reviewer #1: The manuscript has been revised well by the authors. The present edition is recommended to be accepted by the journal Plos One. However, here are two concerns for the authors. First, in line 105-line 111, some conjunctions need to be added to make these sentences more coherent. Second, the chemical structures of Luteolin and Erastin should be provided in the drawn graph using the same color to enhance the professional level.

Reviewer #2: (No Response)

7. PLOS authors have the option to publish the peer review history of their article (what does this mean?). If published, this will include your full peer review and any attached files.

Reviewer #1: No

Reviewer #2: No

---

## [Editor Report · Acceptance letter]

3 Oct 2024

PONE-D-24-25996R1 

PLOS ONE

Dear Dr. Xiao, 

I'm pleased to inform you that your manuscript has been deemed suitable for publication in PLOS ONE. Congratulations! Your manuscript is now being handed over to our production team.

Kind regards, 

on behalf of

Dr. Jian Hao 

Academic Editor

PLOS ONE